# *TPD1-like* Gene as a Suitable Marker for Early Sex Determination in Date Palm (*Phoenix dactylifera* L.)

**DOI:** 10.3390/genes14040907

**Published:** 2023-04-13

**Authors:** Plosha Khanum, Asif Ali Khan, Iqrar Ahmad Khan, Abdul Ghaffar, Zulqurnain Khan

**Affiliations:** 1Institute of Plant Breeding and Biotechnology, MNS-University of Agriculture, Multan 60000, Pakistan; 2Institute of Horticultural Sciences, University of Agriculture, Faisalabad 38040, Pakistan; 3Department of Agronomy, MNS-University of Agriculture, Multan 60000, Pakistan

**Keywords:** Ajwa, Amber, Medjool, suckers, seedling, protein, gene sequence, polymerase chain reaction

## Abstract

Date palm (*Phoenix dactylifera* L.) is a considerably beneficial and economically profitable fruit crop. Female date palm plants produce fruit that is rich in fiber and sugar. Date palm is propagated by two means: suckers and seed. The propagation of date palm through seeds is very necessary for germplasm conservation and breeding. The late reproductive age (4–5 years) and dioecious nature of date palm make genetic improvement and breeding difficult. Early sex determination is the only way to improve breeding by selecting experimental male and female plants at the seedling stage. The primers for *Tapetum Determinant 1* (*TPD1-like*) were designed using Amplify software. The DNA amplification of selected date palm suckers of three genotypes (Ajwa, Amber, and Medjool) was observed through PCR. Expression profiling of selected genotypes was carried out through semi-q PCR and RT-PCR by using the cDNA of suckers and unknown seedlings. Different in silico analyses were performed for the gene and protein characterization and identification of cis-acting elements in the promoter region. The promoter was identified along with the protein’s properties and functionality. The expression of *TPD1-like* gene was found in the leaves of three selected genotypes of male sucker and in some plants of selected unknown seedlings that are considered male plants, and no expression was observed in female suckers and unknown seedlings that are considered female plants. The findings suggested that the *TPD1-like* gene has the potential for sex differentiation at the seedling stage, as the *TPD1-like* gene is essential to the specialization of tapetal cells and plays a critical role in plant reproduction.

## 1. Introduction

Date palm (*P. dactylifera*) is a dioecious, monocotyledonous fruit crop belonging to the family Arecaceae [1]. There are 14 dioecious species of *Phoenix*, but only date palm produces fruit [2]. It is cultivated in many regions of the world, particularly in North Africa, the Middle East and South Asia. It is also called the “tree of life” due to its ability to grow in harsh weather conditions that are not fit for growing other crops. Date palm is extensively grown in dry areas because it can tolerate drought and salinity, and can grow well in alkaline soils [3,4]. The fruit of date palm is a main source of macro- and micronutrients, providing nutritional fiber, soluble sugars, and an ample supply of lipids, proteins and carbohydrates [5,6]. Certain varieties such as Ajwa, Medjool, and Amber are considered to have a high value due to unique characteristics such as their taste, size and texture. These varieties are often in high demand and command a premium price in the market. This can result in a significantly higher income for farmers who grow these varieties compared to those who grow ordinary varieties [7,8].

Seed propagation is required to preserve the genetic variation that promotes date palm breeding and the development of new, improved cultivars. Its breeding is limited due to the need to determine sex at an early stage [9]. Sex identification is an important step in date palm cultivation, as it helps to identify the gender of an individual plant before planting the seedlings. Early-sex identification is used to select a balanced ratio of male and female plants for date palm orchards. By planting productive female plants, growers can reduce the need for the costly maintenance and pruning of unproductive date palm [10,11]. Several morphological, biochemical, and molecular characteristics can be used to identify the sex of dioecious plants at an early stage of development. It is often assumed that the percentage of male and female seedlings in the progeny is equal, which supports the hypothesis that sex in date palm is hereditarily determined [12].

Male and female date palm plants of the same variety can exhibit morphological differences. In general, male date palm plants tend to have a larger leaf length than female plants. However, these morphological differences can be influenced by environmental factor such as soil conditions, water availability, and temperature [13]. The biochemical analysis of date palm leaves does not provide much information about the sex identification of immature plants. The sex of date palm is determined genetically Cytological studies such as chromomycine A3 (CMA3) staining have also been used for sex determination, which depict sex as a genetic factor. The staining technique can reveal differences in the structure and composition of chromosomes, which can help to identify the sex [14]. Over the past decade, various biomarkers, including AFLP, RAPD and microsatellite, have been used to differentiate the sex of date palm plants [15]. 

Gene expression profiling is the most significant and effective technique used to determine an organism’s gene function. In plant biology, a variety of expression profiling techniques have been utilized, and technological development is still ongoing. Additionally, hybridization, sequencing, and polymerase chain reaction (PCR) are three key methods that support these widespread techniques and practices [16]. Real-time polymerase chain reaction (RT-PCR) is one of the best techniques used to evaluate different levels of gene expression. When a certain gene is amplified using gene-specific primers, the fluorescent signals allow for real-time measurements of the target gene level [17]. It has been reported that the *TPD1* gene is required for the development of tapetal cells in *Arabidopsis* anther. Analysis of the male-sterile mutant, *tpd1* showed that the functional interruption of *TPD1* caused the precursors of tapetal cells to differentiate and develop into microsporocytes instead of tapetum and led to the formation of extra microsporocytes, and tapetum was found to be absent in developing anthers. *TPD1* plays an important role in the differentiation of tapetal cells, possibly in coordination with the EMS1/EXS gene product, a Leu-rich repeat receptor protein kinase. It was observed that *TPD1* is initially expressed in leaves and young seedlings [18].

The *TPD1-like* expression in leaves makes it possible to differentiate male and female plants from unknown seedlings, which would be advantageous for early sex identification in date palm [19]. In the current study, we used PCR-based techniques to perform expression profiling of the *TPD1-like* gene for sex determination in specific date palm cultivars. The objective of the study was to find a reliable, stable and durable technique for sex determination in date palm to assist farmers and breeders. 

## 2. Materials and Methods

### 2.1. Plant Material

The leaf samples from suckers of both female and male date palm plants and unknown seedlings of three selected genotypes (Ajwa, Amber, and Medjool) were collected from Horticulture Research Institute, Bahawalpur, Pakistan to evaluate the differential expression of certain sex-determining genes. Date palm leaf tissues from the studied genotypes were thoroughly cleaned in RNase-free water, then dried and frozen for further extraction of RNA and DNA.

### 2.2. DNA Sequence Retrieval and Primer Designing

The *TPD1-like* mRNA sequence of *Phoenix dactylifera* was downloaded from the online portal of the NCBI database with accession number XM-026801684.2 (https://www.ncbi.nlm.nih.gov/ accessed on 11 March 2022). The nucleotide sequence of *TPD1-like* was translated into amino acid sequences through the EXPASY tool (https://web.expasy.org/translate/ accessed on 11 March 2022). PCR primers for *TPD1-like* gene were designed using the software Amplify version 1.7.0. Primer sequences are shown in (Table 1).

### 2.3. Extraction of DNA

The leaf tissue of 300 mg frozen date palm cultivars was ground in 700 μL CTAB extraction buffer until a homogenous mixture. The sample was shifted to a 1.5 mL Eppendorf tube and 50 μL of 2-β-mercaptoethanol was added. This mixture was incubated in water bath for 15 min at 65 °C. Then, the incubated sample was centrifuged at 14,000 rpm for ten minutes. Then, the supernatant was separated and shifted to another clean microfuge tube, and 500 μL of chloroform: Isoamyl Alcohol (24:1) was added and mixed by gentle inversion for 20 min. After mixing, the tube was centrifuged at 14,000 rpm for 10 min. The supernatant was transferred into a new tube, then 500 μL of cold absolute isopropanol was added and mixed carefully by gently inversing the Eppendorf tube. The mixture was incubated for 2 h at −20 °C. The tube was centrifuged at 12,000 rpm for 10 min, and the supernatant was discarded. The DNA pellet was washed twice with 100 μL ice cold 75% ethanol with subsequent centrifugation at 5000 rpm for 5 min. The DNA pellet was air dried and resuspended in 30 μL of distilled water and was stored at −20 °C [20].

### 2.4. RNA Extraction and cDNA Synthesis

Leaf samples kept in liquid nitrogen were crushed to powder for total RNA extraction. RNA extraction of selected samples was performed using TRIzol method [21]. The sample was ground in liquid nitrogen; then, 500 μL of RNA reagent was added and kept on ice for 5 min and spun for 5 min. Then, 500 μL NaCl (5M) + 300 μL chloroform was added to supernatant. Then, 200 μL of chilled isopropanol was added to the solution and held on ice for 10 min. The supernatant was discarded after a 10 min spin and the pellet was resuspended in 200 μL of 70% ethanol and spun for 1–2 min. The liquid was removed by rotating the pellet for 5–6 s while retaining it inside the tube. The pellet was held to dry for 10–15 min. Finally, 30 μL of DEPC water was added to dissolve the pellet. The isolated RNA was used to synthesise cDNA using the cDNA synthesis kit (Thermo Scientific Inc., Waltham, MA, USA) following the manufacturer’s protocols.

### 2.5. Polymerase Chain Reaction

The PCR was conducted using specific primers designed for *TPD1-like* gene. DreamTaq Green PCR Master Mix (2X) (Thermo Scientific Inc., Waltham, MA, USA) was used to prepare PCR reaction following the manufacturer’s protocol. The PCR was performed under the following conditions: initial denaturation at 95 °C for 3 min, followed by 30 cycles of 95 °C for 1 min, 55 °C for 1 min, and 72 °C for 50 s, with final extension at 72 °C for 10 min [22]. Amplicons were analyzed on 1.5% agarose gel using 1× TAE buffer. The integrity of gel bands was checked under UV light using the gel documentation system.

### 2.6. Semi qPCR Analysis of TPD1-like Gene

A reaction mixture (20 µL) of semi-quantitative PCR contained 1 µL template (cDNA reaction product), 1.6 µL MgCl_2_, 2 µL PCR buffer solution, 1 µL dNTP’s mixture, 0.2 µL DNA Taq Polymerase, 1 µL of each primer and 13.2 µL PCR-grade water. PCR program and further procedures were similar to those in Section 2.5.

### 2.7. Real-Time PCR Analysis of TPD1-like Gene

Real-time PCR was performed using real-time system (Bio-Rad Laboratories, Inc., Hercules, CA, USA) with SYBR Green as the reaction mixture, following the manufacturer’s instructions. The reaction mixture (20 µL) contains 5 µL SYBR Green, 0.5 µL of each primer (forward and reverse), 1 µL template of cDNA and 13 µL PCR-grade water. Conditions for amplification were as follows: initial denaturation for 8 min at 95 °C; 36 cycles of quantification consisting of denaturation step for 45 s at 95 °C, annealing for 30 s at 60 °C and extension for 30 s at 72 °C. Glyceraldehyde 3-phosphate dehydrogenase (GAPDH) was used as an internal control gene for the relative expression analysis. Expression was calculated using the 2^−ΔΔCt^ method [23]. 

### 2.8. In Silico Analysis of TPD1-like Gene

#### 2.8.1. Functional Annotation of Protein 

An amino acid sequence of *TPD1****-****like* protein was analyzed using various bioinformatics tools, including TMHMM (https://services.healthtech.dtu.dk/service.php?DeepTMHMM accessed on 31 March 2022), PROTTER (http://wlab.ethz.ch/protter/ accessed on 31 March 2022), SignalP (https://services.healthtech.dtu.dk/service.php?SignalP-5.0 accessed on 31 March 2022) and NetPhos (http://cbs.dtu.dk/services/NetPhos/ accessed on 31 March 2022), to evaluate biochemical and physical characteristics.

#### 2.8.2. Subcellular Localization and Signal Peptide Determination

The SignalP (https://services.healthtech.dtu.dk/service.php?SignalP-5.0 accessed on 1 April 2022) database, which predicts the existence of signal peptides and the locations of their cleavage sites in proteins, was used to determine the position of signal sequence in *TPD1-like* gene [24].

#### 2.8.3. Protein Structure Analysis

The STRING database was used (https://string-db.org accessed on 1 April 2022) to integrate all known and predicted associations between proteins. These connections consist of direct (physical) and indirect (functional) interactions discovered by quantitative modelling, the information conveyed between species, and interactions discovered in other (primary) databases.

#### 2.8.4. Glycosylation Sites Prediction

The web servers GlycoEP (http://www.imtech.res.in/raghava/glycoep accessed on 1 April 2022) and NetNGly (http://www.cbs.dtu.dk/services/NetNGlyc/ accessed on 1 April 2022) were used to more accurately evaluate N-linked, O-linked, and C-linked glycosides in eukaryotic glycoproteins.

#### 2.8.5. Acetylation and Phosphorylation Sites

NetPhos 3.1 (http://www.cbs.dtu.dk/services/NetPhos/ accessed on 5 April 2022) webtoolwas used to examine acetylation and phosphorylation sites. NetAcet 1.0 server (http://www.cbs.dtu.dk/services/NetAcet/ accessed on 5 April 2022) was also used to examine protein acetylation and phosphorylation sites. This included both generic and kinase-specific protein prediction. 

#### 2.8.6. Sequence Analysis of Promoter

The promoter date palm *TPD1-like* gene (XM_026801684.1) was taken from NCBI (http://www.ncbi.nlm.nih.gov/ accessed on 15 April 2022). In BLAST searches, the BLASTx was carried out on the upstream regulatory regions of selected genes. The start codon of the *TPD1* gene was identified and, using several bioinformatics tools, an upstream gene region of about 2.5 kb was evaluated. The chosen sequence was shown to be a non-coding region. About 2 kb of the of *TPD1*-promoter region was identified for isolation and investigated to confirm the noncoding sequences.

Each promoter’s transcription initiation site was located via the BDGP (https://www.fruitfly.org/seq_tools/promoter.html accessed on 15 April 2022) webtool. PlantCARE and plantPAN can be used to identify cis-regulatory elements (http://bioinformatics.psb.ugent.be/webtools/plantcare/html/ accessed on 15 April 2022). PlantPAN 3.0 (http://plantpan.itps.ncku.edu.tw/promoter.php accessed on 15 April 2022) databases were used. Other important regulatory features in the promoter sequence, such as CpG islands and tandem repeats, were also determined by PlantPAN.

## 3. Results

### 3.1. TPD1-like Gene Amplification on DNA of Selected Date Palm Suckers

PCR amplification was carried out using a primer designed from the *TPD1-like* gene (Table 1) on the DNA of selected suckers whose sex was known (male and female plants) to confirm its presence in date palm. The *TPD1-like* gene was amplified in the DNA obtained from the leaves of male and female date palm plants of three selected genotypes, confirming its presence in date palm (Figure 1). 

### 3.2. Semi qPCR Analysis of TPDI-like Gene on the Leaves of Date Palm Suckers

To validate the *TPD1-like* gene for early sex determination at seedling stage in date palm, its expression pattern was determined in the leaves of date palm suckers whose sex was known. It was shown to have varied expression in the leaves of three male genotypes (Ajwa, Amber and Medjool) and showed no expression in the leaves of these female genotypes. The semi-qPCR results were analyzed using ImageJ software [25] to generate a heat map. The heat map generated from this analysis can provide valuable information on the expression of the *TPD1-like* gene in the leaves of selected male and female genotypes through varied color intensity. The dark green color denoted higher expression and orange color denoted no expression (Figure 2). 

### 3.3. Real-Time PCR Analysis of TPD1-like Gene on the Leaves of Date Palm Suckers

From semi-qPCR, the selected leaf tissue of date palm was further analyzed through qPCR, and an expression graph was made that showed that the *TPD1-like* gene was involved in tapetum development and only expressed in male date palm leaves. A higher relative expression was noted in male Medjool plants as compared to male Ajwa and Amber male and no relative expression was observed in female plants of the selected genotypes (Figure 3).

### 3.4. Semi-qPCR Analysis of TPD1-like Gene on the Leaves of Unknown Seedling of Date Palm

The unknown seedlings of Amber, Medjool, and Ajwa were identified on the basis of *TPD1-like* expression analysis. Male plants showed an enhanced expression of *TPD1-like* while female plants of all selected genotypes showed no expression. The heat map generated for unknown date palm seedlings showed varied color expression in selected unknown seedlings of Ajwa, Amber and Medjool plants. In the heat map, a light green to green color showed low–high expression, and orange showed no expression. In the case of Ajwa, out of fifteen seedlings 1, 8, 9, 10, 11, 12, 13 and 14 showed a varied color expression from light green to green for *TPD1-like* and were considered male plants, while seedlings 2, 3, 4, 5, 6, 7 and 15, showed an orange color and were recorded as female plants. Similarly, unknown Amber seedlings showed a varied color intensity for *TPD1-like*; plant numbers 3, 7, 9, 10, 11, 12 and 14 were screened as male, showing a light green and green color, while plants with numbers 1, 2, 4, 5, 6, 8, 13 and 15 had no expression, showing an orange color, and were assumed to be female plants. Plants 1, 2, 5, 6, and 7 for Medjool seedlings showed an orange color and were female while the rest of the plants showed a light green to green color and were considered male plants (Figure 4). 

### 3.5. Real-Time PCR Analysis of TPD1-like Gene on the Leaves of Unknown Seedling of Date Palm

The relative expression of selected unknown seedlings was recorded. Ajwa seedlings with the number 1, 8, 9, 10, 11, 12, 13, 14 showed some expression and were considered male plants, while the rest of the unknown seedlings among the fifteen plants were female plants showing no relative expression (Figure 5A). Unknown Amber seedlings 3, 7, 9, 10, 11, 12 and 14 showed relative expression and were considered male plants (Figure 5B). The Medjool unknown seedlings 3, 4, 8, 9, 10, 11, 12, 13, 14, 15 showed relative expression and were considered male plants, while the rest of the unknown seedlings were considered as female plants showing no expression (Figure 5C). 

### 3.6. In Silico Analysis of TPD1-like Gene

#### 3.6.1. Coding Sequence of *TPD1-like* Gene 

BLASTp was carried out for the sequence identity of *TPD1-like* genes. EXPASY (https://web.expasy.org/translate/ accessed on 11 March 2022) was used to translate the *TPD1-like* gene sequence using six reading frames (Figure 6 and Figure 7).

#### 3.6.2. Subcellular Localization

SignalP was used to predict signal peptides from amino acid sequences. Proteins with signal peptides are targeted to the secretory pathway but are not necessarily secreted. The neural networks in SignalP generated three output scores (C-score, Y-score and S-score) for each position in the input sequence. The findings demonstrate that a similar protein lacks a signal peptide. The output scores and their positions were graphically represented. There is no peptide signal when the C, S, and Y score values are less than 0.45. The sequence from the upstream region of the coding sequence contains a signal peptide with a value of 0.0009 (Sec/SPI), indicating that the signal peptide is located outside the membrane (Figure 8).

#### 3.6.3. Signal Peptide Analysis

The TMHMM membrane protein topology prediction method describes transmembrane helices and various proteins embedded in the cellular membrane. TMHMM analysis showed that *TPD1* protein had no transmembrane proteins, as shown in (Figure 9). The prediction provides the most likely position and direction of TM helices in the sequence. The plot represents the possibilities of the protein both inside and outside of the trans membrane helix.

#### 3.6.4. Acetylation and Phosphorylation Sites

The probable N- phosphorylation sites in protein sequence are predicted using the NetPhos 3.1 server. Phosphorylation is an effective mechanism through which protein activity can be altered once established. Netphos result shows the phosphorylation sites in the *TPD1-like* protein sequence with threshold value 0.5 (Figure 10). Netphos analysis showed that protein has many phosphorylation sites.

#### 3.6.5. Promoter Analysis of *TPD1-like* Gene

A promoter analysis of the *TPD1-like* gene showed predicted transcription initiation site (TIS) with probabilities of 88, 96, 98, 80 and percentages at positions of 33, 150, 209, 294 and 746, respectively. TIS is shown in bold letters in all sequences (Figure 11).

#### 3.6.6. BLASTx Analysis of *TPD1-like* Promoter Sequence

BLASTx results show that the promoter sequence has no similarity with the coding sequence or protein, as shown in Figure 12.

#### 3.6.7. PLANTPAN Analysis of Promoter

PLANTPAN analysis shows various transcription factor binding sites in the promoter region of *TPD1-like* promoter (Figure 13).

#### 3.6.8. Promoter Cis-Acting Regulatory Elements’ Analysis

Using the PLANTCARE database, other cis-regulatory elements were also discovered to be present in the *TPD1* gene’s promoter. PlantCARE analysis shows the expression of different regulatory elements of *TPD1-like*. These regulatory elements include AFI, ARE, TCA element, p-box, CCAATT-box, TATA-box, CAAT-box, and TATC-box, as shown in (Figure 14).

The full-length *TPD1-like* promoter showed binding sites for various transcription factors (Table 2). The full *TPD1* promoter fragment containing core promoter elements, including a CAAT box with consensus sequence CAAAT and CAAT, were also reported in *Pisum sativum* and *Nicotiana glutinosa.* Some light-inducible motifs (G box, AE box, sp1 box) and enhancer elements (CGTCA-motif) were found through bioinformatics analysis.

## 4. Discussion

The sex identification of date palm seedlings is difficult to distinguish before the first flowering stage. Different morphological parameters plant height, leaf length, leaf width and number of leaves have been used to identify the sex of dioecious plants, but these parameters are environmentally dependent. Biochemical studies provide little information about the identification of sex in date palm plants [26]. Molecular techniques offer potential for sex determination as they are stable and independent of age and environment [27]. In the current study, it was found that the *TPD1-like* gene has the potential for early sex determination in different date palm genotypes. Our research suggests that this gene could be a successful approach to determine the sex of date palms at the early seedling stage. Ajwa, Amber and Medjool male plants showed enhanced expression of *TPD1-like.* It was found that female date palm suckers and seedlings of these genotypes did not express the *TPD1-like* gene, which was only expressed in the leaves of male plants. Based on the relative expression of *TPD1-like*, unknown date palm seedlings were differentiated as male and female plants. The *TPD1-like* was markedly expressed in *Arabidopsis thaliana*’s leaves, seedlings, flowering buds, and anthers. *TPD1-like* serves as a ligand for EMS1 receptor kinase to signal cell fate determination during plant sexual reproduction. Its ectopic expression causes the abnormal differentiation of somatic and reproductive cells in anthers. Torres et al. [28] proposed two single copy genes, *CYP-703* and *GPAT3,* the were present in males, involved in male flower development, and absent in females. Our approach to the gene-specific identification of date palm plants at the seedling stage represents a significant success. Transcription factors are proteins that bind to certain DNA sequences to regulate the transcriptional level of gene expression. Therefore, an important aspect of the functional analysis of transcription factors is predicting the DNA-binding motifs. The precise control of gene expression depends on the presence of cis-regulatory elements, and genomic sequences in gene-promoter regions to which transcription factors bind. Due to the central role that these transcription factors play during development, it is necessary to identify anther-development-specific genes that produce transcription factors [29]. Several studies have been proposed regarding the utilization of molecular markers for distinguishing between male and female date palm cultivars, including that of Bekheet and Hanafy [30], Al-Dous et al. [31], Al-Mahmoud et al. [32], Elmeer and Mattat [33], Zhao et al. [34], and Cherif et al. [35]. However, the accuracy of discrimination between sex significantly varies; they are more sensitive and require careful handling and a greater number of repeated experiments. Wang et al. [36] used an SSR marker to differentiate sex in date palm using the qPCR technique. Intha and Chaiprasart [37] reported DNA markers for sex identification in date palm that could differentiate male and female seedlings. Khan et al. [38] proposed reliable and authentic spectroscopic methods that used chemo-informatics regression analysis to discriminate among male and female plants. Our gene-based sex identification of date palm seedlings is a more efficient and time-saving technique. It was found that plant MYB proteins have an impact on a number of developmental processes, as well as signaling and secondary metabolisms [39]. Anthers and pollen contain several genes that could encode transcription factors containing MYB-related genes specified to tapetum from Arabidopsis [40]. A similar MYB motif is identified in the *TPD1-like* promoter analysis, demonstrating its significance in the development of male characters. The expression profiling of genes is a major tool to identify sex at the seedling stage. A few studies provide evidence for the expression of the *TPD1-like* gene in male plants, which supports the idea that this gene is involved in the development of the male character. It was observed that modifying the *TPD1-like* gene in the T-DNA causes metabolic and structural changes, and even male sterility [41].

## 5. Conclusions

The current research is an inclusive study of the identification of sex in date palm using *TPD1-like* gene expression analysis. One-year-old unknown date palm seedlings were identified as male and female plants on the basis of the expression of the *TPD1-like* gene. These findings would support effective genetic resource management, selection, and breeding strategies for date palm. It was concluded that the *TPD1-like* gene is a promising gene to identify the sex of date palm.

## Figures and Tables

**Figure 1 genes-14-00907-f001:**
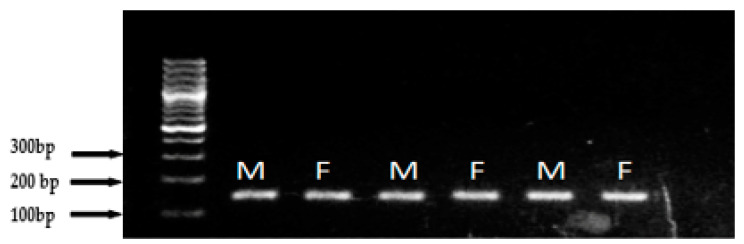
*TPD1-like* gene amplification in leaves using date palm suckers’ DNA as template. PCR product was run on 1.5% agarose gel using a 100 bp ladder (Thermo Scientific Inc., Waltham, MA, USA). A band of 152 bps was observed on the gel. M representing male plants while F representing female plants.

**Figure 2 genes-14-00907-f002:**
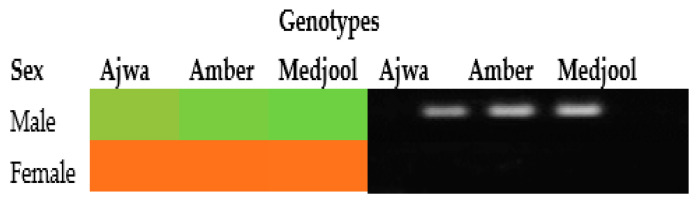
*TPD1-like* amplification in the leaves using date palm sucker cDNA as a template. PCR product was run on 1.5% agarose gel using a 100 bp ladder (Thermo Scientific Inc., Waltham, MA, USA). The heat map indicated the expression intensity: green for high expression, light green for low expression and orange for no expression.

**Figure 3 genes-14-00907-f003:**
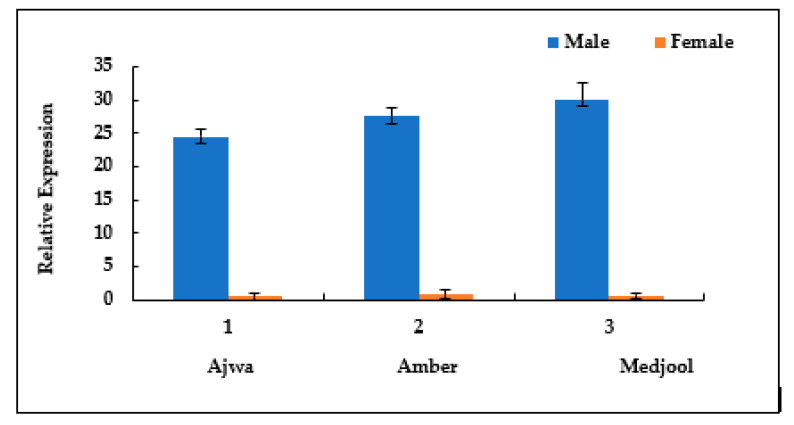
Comparative *TPD1-like* gene expression in the leaves of selected date palm suckers. Error bar indicates the ±SE of three replicates.

**Figure 4 genes-14-00907-f004:**
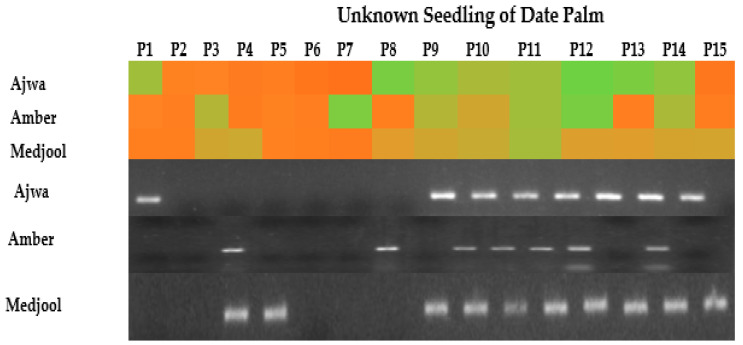
*TPD1-like* gene amplification in leaves using date palm seedling cDNA as template. PCR product was run on 1.5% agarose gel, using a 100 bp ladder (Thermo Scientific Inc., Waltham, MA, USA). The heat map indicated the expression intensity: green for higher expression, light green for low expression and orange for no expression.

**Figure 5 genes-14-00907-f005:**
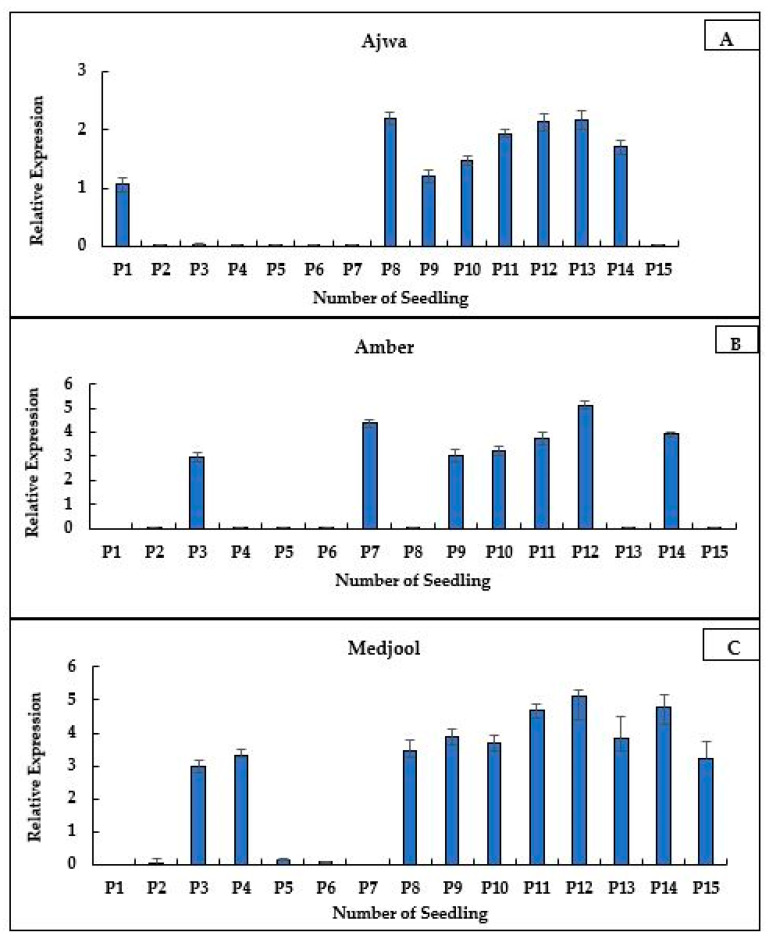
Expression profiling of *TPD1-like* in unknown seedlings of date palm. Error bar indicates ±SE of three experimental replicates. (**A**) is showing expression of *TPD1-like* gene in Ajwa, (**B**) is showing expression of *TPD1-like* gene in Amber and (**C**) is showing expression of *TPD1-like* gene in Medjool.

**Figure 6 genes-14-00907-f006:**
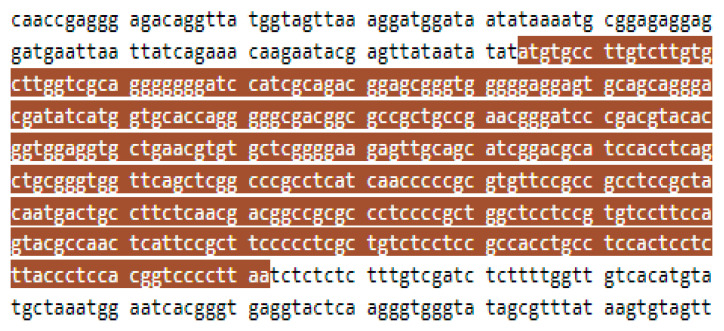
The coding sequence of *TPD1-like* gene (XM_026801684.1) is highlighted.

**Figure 7 genes-14-00907-f007:**
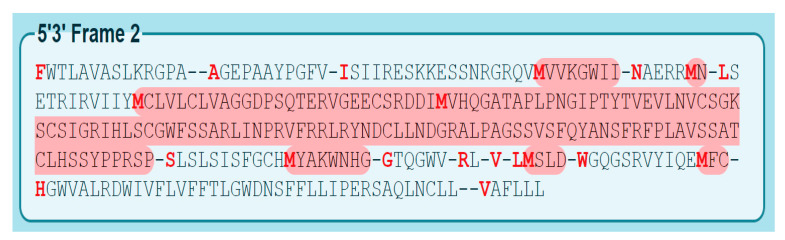
Translated protein sequence of *TPD 1-like* gene containing 132 amino acids (highlighted).

**Figure 8 genes-14-00907-f008:**
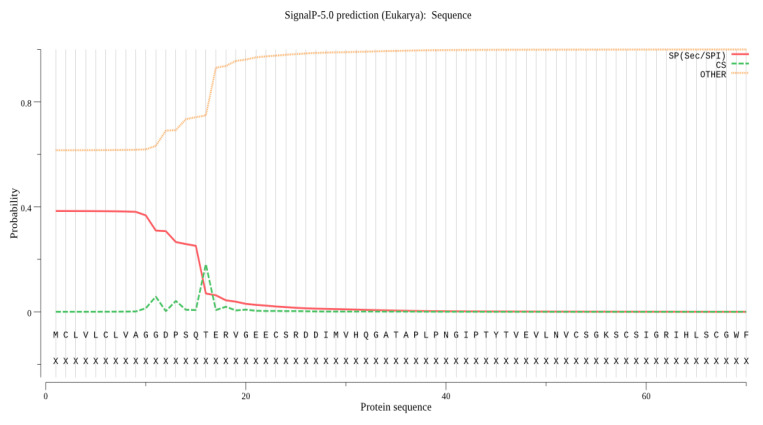
Graphically signal peptide representation of *TPD1-like* protein.

**Figure 9 genes-14-00907-f009:**
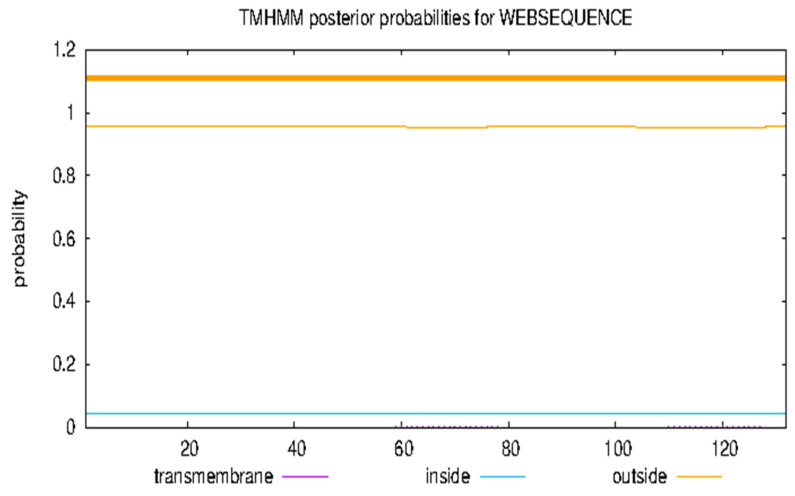
TransMembrane prediction using Hidden Markov Models (TMHMM) of *TPD1-like* protein.

**Figure 10 genes-14-00907-f010:**
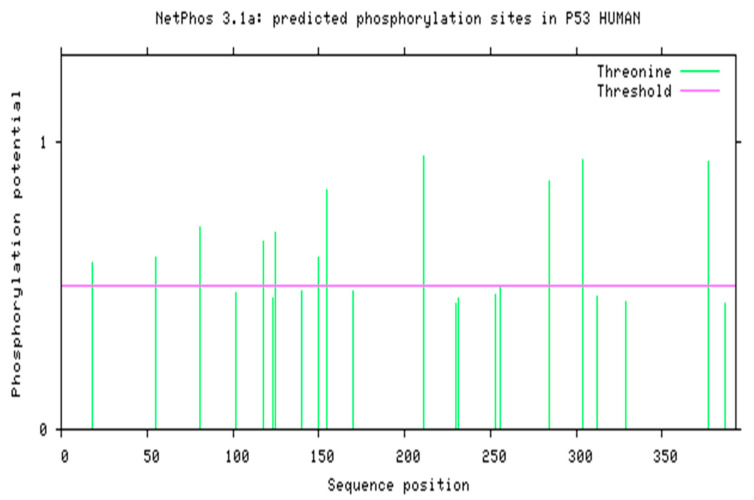
*TPD1-like* protein’s predicted phosphorylation locations.

**Figure 11 genes-14-00907-f011:**
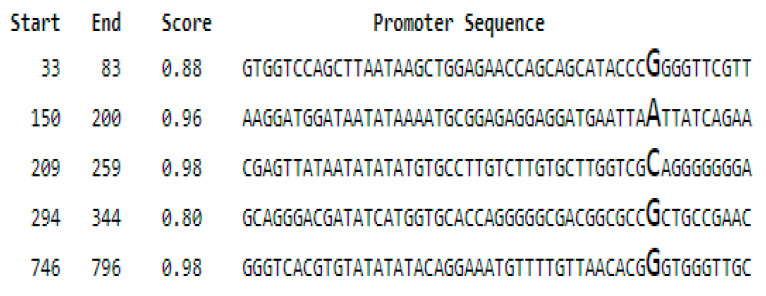
Predicted transcription start site in *TPD1* promoter with % probability.

**Figure 12 genes-14-00907-f012:**
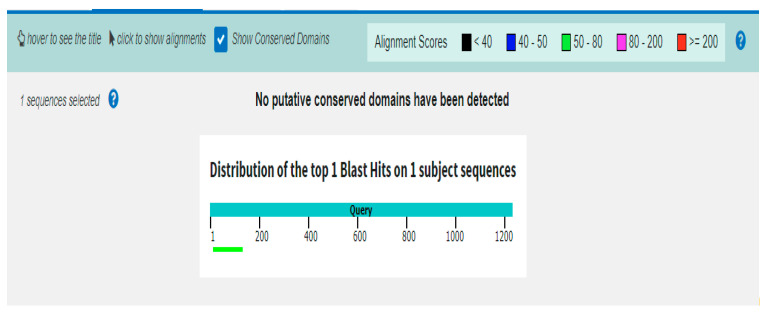
BLASTx analysis of *TPD1* promoter showing no coding region and conserved domains.

**Figure 13 genes-14-00907-f013:**
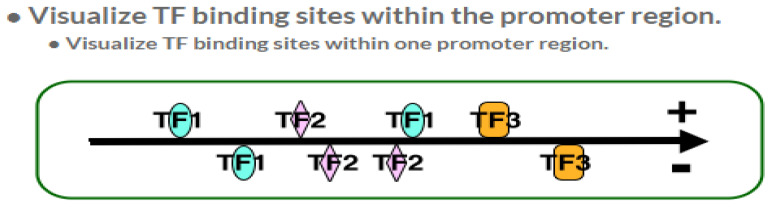
Transcriptional factor binding sites in *TPD1*-promoter region.

**Figure 14 genes-14-00907-f014:**
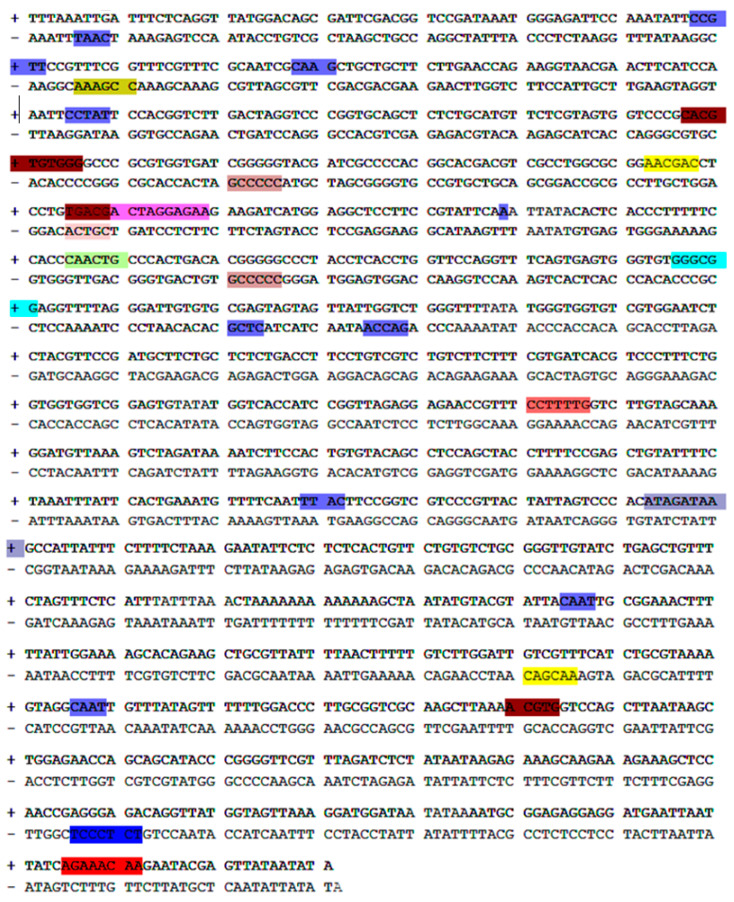
Promoter sequence of *TPD1* gene; important motifs are highlighted in various colors.

**Table 1 genes-14-00907-t001:** Primer used for selected date palm suckers and unknown seedlings for PCR amplification.

Gene Abbreviation	Name of Gene and Accession No.	Primer Length (bp)	Forward (5′-3′)	Reverse (5′-3′)
*Tapetum Determinant 1-Like*	*PdTPD1-like* (>XM_026801684.2)	152	ATCGGACGCATCCACCTCAG	GAGTTGGCGTACTGGAAGGACA

**Table 2 genes-14-00907-t002:** Site name, position, sequence and function of primers of organisms.

Site Name	Organism	Position Standard	Sequence	Function
ABRE	*A. thaliana*	+1030	ACGTG	cis-acting component involved in abscisic acid responsiveness
ABRE	*Hordeum vulgare*	+205	CGCACGTGTC	cis-acting component involved in abscisic acid responsiveness
ABRE	*A. thaliana*	+207	CACGTG	cis-acting component involved in abscisic acid responsiveness
ABRE	*A. thaliana*	+208	ACGTG	cis-acting element involved in the abscisic acid responsiveness
ABRE	*A. thaliana*	−547	ACGTG	cis-acting component involved in abscisic acid responsiveness
AE-box	*A. thaliana*	+1195	AGAAACAA	part of a module for light response
CAAT-box	*P. sativum*	+60	CAAAT	common cis-acting element in promoter and enhancer regions
CAAT-box	*P. sativum*	+139	CAAAT	cis-acting component involved in abscisic acid responsiveness
CAAT-box	*N. glutinosa*	+725	CAAT	common cis-acting element in promoter and enhancer regions
CGTCA-motif	*H. vulgare*	−285	CGTCA	cis-acting regulatory element involved in the MeJA-responsiveness
G-box	*A. thaliana*	+207	CACGTG	cis-acting regulatory element involved in light responsiveness
G-box	*Zea mays*	−479	CACGAC	cis-acting regulatory element involved in light responsiveness
GA-motif	*A. thaliana*	+763	ATAGATAA	part of a light-responsive element
GC-motif	*Z. mays*	−231	CCCCCG	enhancer-like element involved in anoxic specific inducibility
GC-motif	*Z. mays*	−371	CCCCCG	enhancer-like element involved in anoxic specific inducibility
LTR	*H. vulgare*	−76	CCGAAA	cis-acting element involved in low-temperature responsiveness
MBS	*A. thaliana*	+355	CAACTG	MYB binding site involved in drought-inducibility
Myb	*A. thaliana*	+355	CAACTG	Stress inducubility
Myc	*A. thaliana*	−1095	TCTCTTA	Stress inducubility
P-box	*Oryza sativa*	+611	CCTTTTG	gibberellin-responsive element
Sp1	*O. sativa*	+416	GGGCGG	Light-responsive element
TGACG-motif	*H. vulgare*	+285	TGACG	cis-acting regulatory element involved in the MeJA-responsiveness

## Data Availability

The corresponding author can provide access to all the data that was used and examined in the current field investigation.

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
