# Peer review of "TPD1-like Gene as a Suitable Marker for Early Sex Determination in Date Palm (Phoenix dactylifera L.)"

_genes, 2023, doi:10.3390/genes14040907_

Round 1
Reviewer 1 Report
The authors describe the application of TPD1 gene expression as a reliable marker for sex determination in date palms, which are dioecious, economically important, and take 4-5 years to flower. However, there are many relevant points as below, which the authors need to address. The manuscript writing is quite immature and needs drastic improvisation.
1. The authors do not describe the basis of their selection of the TDP1 gene. How did they pick this gene? Was it from previous transcriptome data or any other? However, there is a list of many genes in Table 1 of the manuscript, for which the authors have designed primers, but have not mentioned their purpose in the manuscript.
2. There are many previous reports of molecular marker identification for sex determination in date palms. The authors have not mentioned these in the manuscript. They should have discussed how these are similar/different from TPD1. Some examples include Kharb and Mitra (Methods Mol Biol. 2017; 1638:199-207. doi: 10.1007/978-1-4939-7159-6_17.), Intha and Chaiprasart (Scientia Horticulturae 2018; 236:251-255), Torres et al. 2021 (https://www.frontiersin.org/articles/10.3389/fpls.2021.634901/full), Al-Qurainy, et al 2018 (https://www.hindawi.com/journals/ijg/2018/3035406/), Torres et al 2018 (https://www.nature.com/articles/s41467-018-06375-y). Other methods used to differentiate gender in date palm [example, Khan et al 2021 (Plants | Free Full-Text | Spectroscopic and Molecular Methods to Differentiate Gender in Immature Date Palm (Phoenix dactylifera L.) (mdpi.com))] could have been included in the discussion.
3. The manuscript is poorly written.
4. The abstract does not convey about the in silico prediction described in the manuscript.
5. The grammatical errors across the manuscript are too many to correct. Some examples include -
Line 12-14: Reframe the sentences ‘Date palm can be grown from seed, but it takes four to five years to reach the first 12 flowering stage that takes too much time to identify the sex of plants. Furthermore, it multiplies slowly through suckers, often serving as a disease carrier.’.
Line 14: ‘….often serving as a disease carrier.’ How? This sentence is convening partial information. Can be deleted.
Line 16: ‘…… identified by applying 15 genes to the leaves…..’ How do you apply genes?
Line 18: ‘…..specialization….’ of what?
Author Response
The authors are thankful to the reviewer for his/her time to read their article for improvement. Valuable suggestions have been appreciated and incorporated in the updated version is attached herewith. The response to comments is given below;
|
Sr. No. |
Reviewer Comments |
Needful Done |
Page No. |
|
1 |
The authors do not describe the basis of their selection of the TDP1 gene. How did they pick this gene? Was it from previous transcriptome data or any other? However, there is a list of many genes in Table 1 of the manuscript, for which the authors have designed primers, but have not mentioned their purpose in the manuscript. |
The authors are thankful to the reviewer for reviewing their article critically for improvement. TPD1 was previously studied in Arabidopsis and it was found that this gene involved in tapetum development which may be used for sex determination added in Abstract section and Introduction section. The mRNA sequence was taken from NCBI, the details are given in the methods. Fifteen primers were designed from selected genes mentioned in (Table 1) and their PCR amplification was tried on DNA and cDNA of three selected date palm genotypes. TPD1 gene showed more promising results and was selected for further studies. Other primers have been excluded from the table. |
Page#1 Line #20-21, Page#2-3 line # 85-94. |
|
2 |
There are many previous reports of molecular marker identification for sex determination in date palms. The authors have not mentioned these in the manuscript. They should have discussed how these are similar/different from TPD1. Some examples include Kharb and Mitra (Methods Mol Biol. 2017; 1638:199-207. doi: 10.1007/978-1-4939-7159-6_17.), Intha and Chaiprasart (Scientia Horticulturae 2018; 236:251-255), Torres et al. 2021 (https://www.frontiersin.org/articles/10.3389/fpls.2021.634901/full), Al-Qurainy, et al 2018 (https://www.hindawi.com/journals/ijg/2018/3035406/), Torres et al 2018 (https://www.nature.com/articles/s41467-018-06375-y). Other methods used to differentiate gender in date palm [example, Khan et al 2021 (Plants | Free Full-Text | Spectroscopic and Molecular Methods to Differentiate Gender in Immature Date Palm (Phoenix dactylifera L.) (mdpi.com))] could have been included in the discussion. |
Authors found reviewer’s suggestions very valuable and quite relevant. The articles have been discussed and cited in the Discussion section |
Page#15 Line#440-461 |
|
3 |
The manuscript is poorly written |
Manuscript has been improved thoroughly |
|
|
4 |
The abstract does not convey about the in-silico prediction described in the manuscript. |
Added in Abstract |
Page# 1 Line # 24-26.
|
|
5 |
The grammatical errors across the manuscript are too many to correct. Some examples include - Line 12-14: Reframe the sentences ‘Date palm can be grown from seed, but it takes four to five years to reach the first 12 flowering stage that takes too much time to identify the sex of plants. Furthermore, it multiplies slowly through suckers, often serving as a disease carrier.’Line 14: ‘often serving as a disease carrier.’ How? This sentence is convening partial information. Can be deleted. Line 16: ‘…… identified by applying 15 genes to the leaves…..’ How do you apply genes? Line 18: ‘….specialization….’ of what? |
All suggested correction has been done in whole manuscript. |
Page #1 Line #16-18. Page #1 Line #14-15. Page #1 Line #30-32 |

Reviewer 2 Report
Expression analysis of TPDI was not clear. Figure 1 described TPD1 gene amplification in leaves using date palm suckers DNA as template on agarose gel; while Figure 2 & Figure 3 showed TPD1 gene expression level in different samples without original results. The relative expression was analyzed by real-time PCR equipment or was calculated based on PCR product on agarose gel or PAGE? No detailed information was provided in Materials and Methods, while “TPD1 expression analysis was examined using cDNA” was mentioned in Results 3.2.
Besides, sex determination in date palm can be detected directly in PAGE by showing distinct bands (For example, Wang et al., 2020; Crop Science, DOI: 10.1002/csc2.20187), while seems not so convincing by showing more or less expression. Especially in Figure 3, no significant difference was showed between P1 and P2. However, they were recorded as male and female date palm suckers (Figure 3 A), respectively.
Author Response
The authors are thankful to the review for critical review and comments on their article which helped in the improvement of the article. All valuable suggestions have been accepted in the updated version of the manuscript. The response to comments is given below;
Reviewer II
|
Reviewer Comments |
Needful Done |
Page No. |
|
Expression analysis of TPDI was not clear. Figure 1 described TPD1 gene amplification in leaves using date palm suckers DNA as template on agarose gel; while Figure 2 & Figure 3 showed TPD1 gene expression level in different samples without original results. The relative expression was analyzed by real-time PCR equipment or was calculated based on PCR product on agarose gel or PAGE? No detailed information was provided in Materials and Methods, while “TPD1 expression analysis was examined using cDNA” was mentioned in Results 3.2.
|
The authors are thankful to the reviewer for reviewing their article critically for improvement. The detail related to PCR analysis have been added in Materials and Methods section.
|
Page no.4, Line # 149-169. |
|
Besides, sex determination in date palm can be detected directly in PAGE by showing distinct bands (For example, Wang et al., 2020; Crop Science, DOI: 10.1002/csc2.20187), while seems not so convincing by showing more or less expression. Especially in Figure 3, no significant difference was showed between P1 and P2. However, they were recorded as male and female date palm suckers (Figure 3 A), respectively |
The detail of results have been added with gel pic and graph according to your king suggestions. |
Page #6-9, Line # 217-281. |

Reviewer 3 Report
Dear Authors
Please see some comments.

Author Response
The authors are thankful to the review for critical review and comments on their article which helped in the improvement of the article. All valuable suggestions have been accepted in the updated version of the manuscript. The response to comments is given below;
Reviewer III
|
Reviewer Comments |
Needful Done |
Page No. |
|
Suggested title: TPD1 gene, as a suitable gene marker to identify male seedlings of date palm cultivars |
The authors are thankful to the reviewer for reviewing their article critically for improvement. The detail related to PCR analysis have been added in Materials and Methods section. TPD1 gene as a suitable marker for early sex determination in date palm (Phoenix dactylifera L.)
|
Page no.1, Line # 2-3. |
|
Please select the keywords that they were not repeated in title of paper. |
Corrected and added according to your kind suggestion. |
Page no.1, Line # 33-34 |
|
Extraction of DNA Reference? |
Added |
Page no.15 Line # 487 |
|
Polymerase Chain Reaction Reference? |
Added |
Page no.15 Line # 489 |

Round 2
Reviewer 1 Report
The authors have provided line numbers in their comments, but these are missing in the manuscript. Please provide the same to ensure proper reviewing.
Author Response
The authors are thankful to the Reviewer for his/her valuable time to review their article and providing suggestion for improvement. All suggestion were taken seriously and accepted accordingly. The page number and line number have been provided in the manuscript and response to the comments. All corrections are green highlighted.
The response to the comments is attached herewith.

Round 3
Reviewer 1 Report
The line numbers described by authors do not correlate with the information in the respective lines in the pdf. As the authors have not numbered the lines in their word file, it is very difficult to locate the information there as well. Looks like some of the corrections have not been incorporated, or, I am unable to locate those in the said line numbers.
Though the authors have re-worked on the abstract, some grammatical errors can still be seen. Some words are fused in the manuscript.
Also, the newly incorporate refrences by the authors do not seem follow the same style during citation.